# On Secrecy Performance of SWIPT Energy-Harvesting Relay Jamming Based Mixed RF-FSO Systems

**Yi Wang** [1,2,*]**, Zhiwu Zhan** [1] **and Zihe Shen** [1]

1 Key Laboratory of Electromagnetic Wave Information Technology and Metrology of Zhejiang Province, College of Information Engineering, China Jiliang University, Hangzhou 310018, China; p20030854044@cjlu.edu.cn (Z.Z.); p21030854041@cjlu.edu.cn (Z.S.)
2 State Key Laboratory of Coal Resources and Safe Mining, China University of Mining and Technology, Xuzhou 221116, China
* Correspondence: wcy16@cjlu.edu.cn

**Abstract:** This paper proposes a simultaneous wireless information and power transfer (SWIPT) energy-harvesting relay jamming based mixed RF/FSO system, and studies its security performance optimization in the presence of an eavesdropper. In this work, the RF and FSO channels experience Nakagami-m fading distribution and Málaga(M) turbulence, respectively. A two-hop decode-and-forward (DF) relay is presented in the system, and the effect of pointing errors is considered. The presence of a nearby single antenna eavesdropper that attempts to eavesdrop on the transmission is also modeled. In order to prevent eavesdropping, the relay introduces the SWIPT structure to control information delivery and wireless energy recharging. The closed expressions of secrecy outage probability (SOP) and average secrecy capacity (ASC) of the mixed RF/FSO system are derived for the above system model. In addition, the closed-form expression of the asymptotic results for SOP and ASC are derived when signal-to-noise ratios at relay and legitimate destinations tend to infinity. The correctness of these expressions is verified using the Monte Carlo method. The influence of various key factors on the safety performance of the system is analyzed by simulations. The results show that the safety performance of the system is considerably improved under good weather conditions as well as by increasing the signal-interference noise ratio, number of interferer antennas, power distribution factor and energy conversion efficiency. This study provides a new system structure and a good theoretical basis for evaluating the physical layer security performance of the mixed RF/FSO system.

**Keywords:** RF/FSO; physical layer security; energy-harvesting relay interference; SWIPT

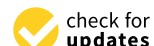



## 1. Introduction

The spectrum scarcity is becoming increasingly evident due to the explosive growth in demand for wireless devices and networks. Therefore, free-space optical communication (FSO), also known as atmospheric optical communication, was proposed [1]. Compared with traditional radio frequency (RF) communication, the FSO has the advantages of free licenses, low cost and high bandwidth. In addition, the FSO technology can effectively solve the "last mile" access request, especially when the communication link is being installed in a complex terrain. Compared to fiber communication, FSO communication is cost-effective, flexible and convenient.

Although the FSO bears many advantages over the RF, it also has a few unavoidable problems. On the one hand, the FSO links are susceptible to atmospheric turbulence caused by random changes in atmospheric temperature and pressure during communication. This causes the intensity of the optical signal transmitted in the atmosphere to fluctuate randomly, resulting in the phenomenon of light intensity flicker, which seriously affects the communication performance of the FSO link. On the other hand, the pointing errors caused by swaying buildings and path loss caused by weather conditions and transmission

path length can also affect the communication performance of the FSO link [2]. In order to solve the aforementioned problems, a new communication model with unique advantages, known as the mixed RF/FSO system is proposed [3,4]. Combining the reliability of RF communication and the security of FSO communication under severe weather conditions, the two communication technologies can fully realize their respective advantages. This can not only improve the overall performance of the communication system, but also effectively expand the communication distance.

In recent years, researchers started studying the performance of the mixed RF/FSO relay systems, proving the feasibility of improving the performance reliability of FSO communication by means of a relay. Considering the broadcast property of a wireless RF channel in a hybrid system, the confidential information in RF communication can be easily acquired by malicious eavesdroppers, which poses serious security risks in communication. Subsequently, the physical layer security (PLS) performance of the mixed RF/FSO system is studied. As the PLS does not depend on any encryption algorithm and has low complexity, only the physical characteristics of a wireless channel can be used to achieve perfectly secure communication. Therefore, the PLS of mixed RF/FSO systems has become a popular research topic.

So far, there is a limited amount of literature that analyzes the PLS of mixed RF/FSO systems. Existing literature studies the influence of partial relay selection and TAS on the PLS of mixed systems. For example, Odeyemi et al. studied the secrecy performance of a partial relay selection based mixed RF/FSO system with outdated channel state information [5]. Lei et al. considered the channel state information outdated of RF and FSO links, analyzed the PLS of multiple input multiple output (MIMO) mixed RF/FSO system, and proposed four transmit antenna selection (TAS) schemes to improve the system's security performance [6]. The influence of relay structure changes on the PLS of mixed RF/FSO systems has also been studied. For example, Pattanayak et al. investigated the PLS performance of a dual hop two-way relaying (TWR) based mixed FSO/RF system [7]. In addition, the PLS of mixed RF/FSO systems is studied with respect to channel fading distribution. For example, Islam et al. considered the RF links undergoing generalized Gamma fading distribution and FSO links undergoing M fading distribution relative to the traditional fading distribution to analyze the secrecy performance of mixed systems [8]. Juel et al. studied the influence of the RF link experiencing the $\alpha$-$\mu$ distribution and the FSO link experiencing the Weibull distribution on the PLS of a system [9].

In addition, there are recent studies on the PLS of energy harvesting wireless networks. Radio frequency energy harvesting wireless networks are mainly divided into simultaneous wireless information and power transfer (SWIPT) and wireless powered communication networks (WPCN). The latter mainly divides wireless communication into power transfer and information transmission phases [10]. For example, Wang et al. investigated the PLS performance of a mixed RF/FSO system in the presence of an eavesdropper under the influence of WPCN-friendly jammers [11]. Compared with the WPCN, the RF signals in a SWIPT structure can not only carry data, but can also be used as a wireless power transmission charging source for energy-limited communication devices [12]. Therefore, the SWIPT reduces the communication time of wireless devices and enables dual use of RF signals.

For example, Lei et al. analyzed the impact of SWIPT and multi-antenna technologies on the security performance [13]. Saber et al. studied the security performance of a mixed RF/FSO system for single-input multiple-output (SIMO) SWIPT [14]. Existing SWIPT studies have concentrated on the possibility of energy harvesting devices acting as possible eavesdroppers, and have not taken advantage of the characteristics of energy harvesting that can interfere with eavesdroppers. In all the aforementioned studies relating to the security performance of the mixed RF/FSO system, the main aim is to improve the information transmission rate from the sender to the legitimate receiver. There are no reports on reducing the eavesdropping receiver rate in the physical layer security transmission technology.

Based on the above discussion, a mixed RF/FSO system based on SWIPT energy-harvesting relay jamming is proposed in this paper. The RF links experience Nakagami-m fading distribution, and FSO links experience M fading distribution. The SWIPT energy collection technology and decoding and forwarding (DF) relay scheme are adopted. The main contributions of this work are as follows: First, the cumulative distribution function (CDF) and the probability density function (PDF) of the signal-interference noise ratio (SINR) for the eavesdropper under the action of energy-harvesting relay jamming are derived. Second, the uniform CDF of the end-to-end signal-to-noise ratio (SNR) of the SIMO communication system under the DF relay scheme is obtained. Third, the closed and asymptotic approximate expressions for the system SOP and the ASC are derived using the CDF and based on the Meijer-G function and the generalized Gauss–Laguerre formula. Fourth, the accuracy of the expression is verified using the Monte Carlo methods. Last, theoretical analysis is used to analyze, the effects of the average SINR of the energy-harvesting relay, the number of interferer antennas, power distribution factor and energy conversion efficiency, and weather conditions on the system security performance via simulations.

## 2. System and Channel Model

As Figure 1 shows, a SWIPT energy-harvesting relay jamming based mixed RF/FSO system is considered. The system consists of a single antenna RF source (S), a multi-antenna DF relay (R) and a single antenna target node (D). The RF link in the S-R segment and the FSO link in the R-D segment undergo Nakagami-m distribution and M distribution, respectively. A single antenna eavesdropper (E) attempting to eavesdrop on the channel transmission is located in the RF link near the energy-harvesting relay. The energy-harvesting relay converts the received information into an optical signal, which is then transmitted through the atmospheric turbulence channel in the presence of pointing errors.

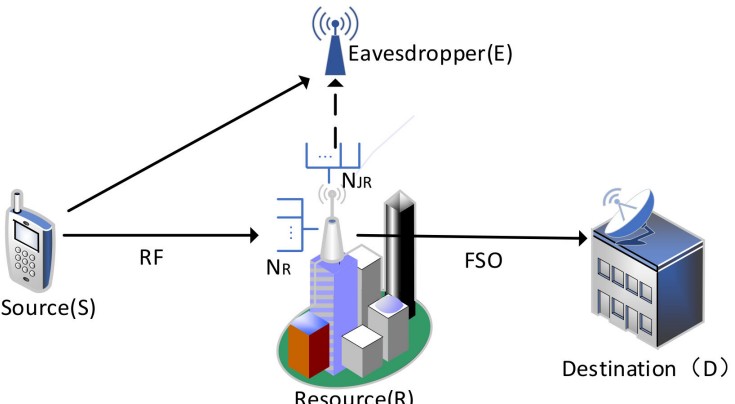

**Figure 1.** A SWIPT Energy-Harvesting relay jamming based Mixed RF/FSO system.

Figure 2 shows the structure model of SWIPT power switching receiver. In the power switching scheme, a power split is installed at the relay receiver, which divides the received wireless signal into two different power streams: (1) One part is transmitted to the rectifier antenna for energy collection, and (2) the other part is converted into baseband signal for information decoding, which can easily realize wireless energy-carrying communication. The tradeoff between energy harvesting and information transfer rate can be achieved by appropriately setting the weights λ between energy and information reception.

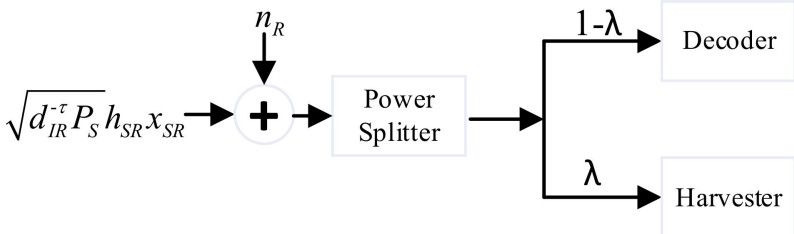

**Figure 2.** SWIPT power switch receiver structure.

The source node $S$ sends a radio signal $x_{SR}$ of power $P_S$ to the energy collector and information receiver, known as a relay. The RF link undergoes a static Nakagami-m distribution, and the signal received at the relay information receiver and energy collector is given as:

$$y_{SR}^{IR} = \sqrt{1-\lambda}\left(\frac{1}{\sqrt{d_{IR}^{\tau}}}\sqrt{P_S}h_{IR}x_{SR} + n_R\right)$$
$$y_{SR}^{EH} = \sqrt{\lambda}\left(\frac{1}{\sqrt{d_{EH}^{\tau}}}\sqrt{P_S}h_{EH}x_{SR} + n_R\right) \tag{1}$$

where $P_S$ is the transmitting power at the transmitter, $x_{SR}$ is the normalized signal of the source node, $h_{IR}$ and $h_{EH}$ are channel coefficients between source node $S$, and information receiver and energy collector respectively, and $d_{IR}$ and $d_{EH}$ represent the distances between $S$, and the information receiver and energy collector, respectively, The path loss exponent is denoted by $\tau$, and $n_R$ denotes additive Gaussian white noise at the interferer with zero mean and variance $\sigma_{SR}^2$.

According to Equation (1), when the noise is ignored, the energy obtained by the energy collector at the relay terminal is expressed as: [15]:

$$E_R = \lambda\zeta\frac{1}{d_{SR}}P_S h_{SR}^2 T \tag{2}$$

where $\zeta$ is the energy conversion efficiency of the energy-harvesting relay to convert the RF signal to direct current (DC), $T$ is the time of energy collection, and $h_{SR}$ is channel coefficient between source node $S$ and the energy-harvesting relay.

As the relay has multiple jamming transmitting antennas $N_J$, it can be gathered through the artificial interference generation method in [16] where the energy-harvesting relay generates a null space orthogonal basis matrix $\mathbf{W}$ of size $N_J \times (N_J - 1)$. The vector $\chi$ has $(N_J - 1)$ independent identically distributed complex Gaussian random elements with normalized variances. The energy-harvesting relay then sends $\mathbf{W}\chi$ as an interference signal. When $N_J > 1$, the signal received at the relay is not affected by the interference signal as it is transmitted to the null space.

The signal received at the eavesdropper under the action of energy-harvesting relay interference can be expressed as follows:

$$y_{SE} = \frac{\sqrt{P_S}}{\sqrt{d_{SE}^{\tau}}}h_{SE}x_{SR} + \frac{\sqrt{P_R}}{\sqrt{d_{RE}^{\tau}}}h_{RE}\frac{\mathbf{W}\chi}{\sqrt{N_J - 1}} + n_E \tag{3}$$

where $h_{SE}$ is the channel coefficient between the source node $S$ and the eavesdropper node, $h_{SR}$ is channel coefficient between the energy-harvesting relay and the eavesdropper $E$, and $n_E$ is the additive Gaussian white noise at the relay with zero mean and $\sigma_E^2$ variance.

### 2.1. RF Channel Model

The interference signal sent by the energy-harvesting relay interference will only affect the eavesdropper. In the communication scene, the SINR of the eavesdropper is as follows:

$$\gamma_E = \frac{\gamma_{SE}}{\gamma_{JE} + 1} \tag{4}$$

where $\gamma_{JE} = \frac{P_R||h_{JE}W||^2}{\sigma_E^2 d_{JE}^\tau(N_J-1)} = \frac{\lambda\zeta P_S N_J ||h_{JE}W||^2|h_{SR}|^2}{d_{SR}^\tau d_{JE}^\tau N_0(N_J-1)}$, $\gamma_{SE} = \frac{P_s|h_{SJ}|^2}{\sigma_E^2 d_{SE}^\tau}$.

All RF links between S-R undergo Nakagami-m fading distribution, and PDF and CDF of the instantaneous SNR $\gamma_\kappa$ of the RF links are given as:

$$f_{\gamma_k}(\gamma) = \frac{1}{\Gamma(m_k N_k)}\left(\frac{m_k}{\Omega_k}\right)^{m_k N_k} \gamma_k^{m_k N_k - 1} \exp\left(-\frac{m_k}{\Omega_k}\gamma_k\right) \tag{5}$$

$$F_{\gamma_k}(\gamma) = 1 - \exp\left(-\frac{m_k}{\Omega_k}\gamma_k\right) \sum_{t=0}^{m_k N_k - 1} \frac{1}{t!}\left(\frac{m_k}{\Omega_k}\gamma_k\right)^t \tag{6}$$

where $\kappa \in \{SR, SE, JE\}$, and the terms $\Omega_{SR} = \frac{P_S \lambda_{SR}}{\sigma_{SR}^2 d_{SR}^\tau}$, $\Omega_{SE} = \frac{P_S \lambda_{SE}}{\sigma_E^2 d_{SE}^\tau}$, and $\Omega_{JE} = \frac{\lambda\zeta P_S \lambda_{JE} N_{JR}}{\sigma_E^2 d_{SR}^\tau d_{JE}^\tau}$ represent the average power channel gain of each channel, respectively.

Using Equations (4)–(6), the CDF of the instantaneous SNR $\gamma_{SJE}$ is obtained as follows [17]:

$$\begin{aligned}F_{\gamma_{SJE}}(\gamma) &= 1 - \sum_{i=0}^{m_{SE}-1} \frac{1}{t!}\left[\frac{m_{SE}}{\Omega_{SE}(N_J-1)}\right]^t \frac{1}{\Gamma(m_{RE}(N_J-1))} \sum_{k=0}^t \binom{t}{k}\left(\frac{\Omega_{RE}}{m_{RE}}\right)^k \gamma^t \\ &\times \exp\left[-\frac{m_{SE}}{\Omega_{SE}(N_J-1)}\gamma\right] G_{1,1}^{1,1}\left[\frac{m_{SE}\Omega_{RE}}{\Omega_{SE}(N_J-1)}\gamma \middle| \begin{matrix} 1-k-m_{RE}(N_J-1) \\ 0 \end{matrix}\right]\end{aligned} \tag{7}$$

The derivative operation is performed on Equation (7) to obtain the following expressing of the PDF:

$$\begin{aligned}f_{\gamma_{SJE}}(\gamma) &= \frac{1}{\Gamma(m_{SE})\Gamma(m_{RE}(N_{JR}-1))}\left[\frac{m_{SE}}{\Omega_{SE}(N_{JR}-1)}\right]^{m_{SE}} \sum_{k=0}^{m_{SE}-1} \binom{m_{SE}-1}{k}\left(\frac{\Omega_{RE}}{m_{RE}}\right)^k \gamma^{m_{SE}-1} \\ &\times \exp\left[-\frac{m_{SE}}{\Omega_{SE}(N_J-1)}\gamma\right] G_{1,1}^{1,1}\left[\frac{m_{SE}\Omega_{JE}}{\Omega_{SE}m_{RE}(N_J-1)}\gamma \middle| \begin{matrix} 1-k-m_{RE}(N_J-1) \\ 0 \end{matrix}\right]\end{aligned} \tag{8}$$

*2.2. FSO Channel Model*

$$f_{\gamma_{RD}}(\gamma) = \frac{g^2 A}{2\gamma_{RD}} \sum_{m=1}^\beta b_m G_{1,3}^{3,0}\left[\frac{B\gamma_{RD}}{\overline{\gamma_{RD}}} \middle| \begin{matrix} 1+g^2 \\ g^2, \alpha, m \end{matrix}\right] \tag{9}$$

$$F_{\gamma_{RD}}(\gamma) = \frac{g^2 A}{2} \sum_{m=1}^\beta b_m G_{2,4}^{3,1}\left[\frac{B\gamma_{RD}}{\overline{\gamma_{RD}}} \middle| \begin{matrix} 1, 1+g^2 \\ g^2, \alpha, m, 0 \end{matrix}\right] \tag{10}$$

where $B = \frac{g^2 \alpha\beta(\xi_g+\Omega')}{(g^2+1)(\xi_g\beta+\Omega')}$, $A = \frac{2\alpha^{\frac{\alpha}{2}}}{\xi_g^{1+\frac{\alpha}{2}}\Gamma(\alpha)}\left(\frac{\xi_g\beta}{\xi_g\beta+\Omega'}\right)^{\beta+\frac{\alpha}{2}}$, $b_m = a_m\left(\frac{\alpha\beta}{\xi_g\beta+\Omega'}\right)^{-((\alpha+m)/2)}$,

$a_m = \binom{\beta-1}{m-1}\frac{(\xi_g\beta+\Omega')^{1-(\frac{m}{2})}}{(m-1)!}\left(\frac{\Omega'}{\xi_g}\right)^{m-1}\left(\frac{\alpha}{\beta}\right)^{\frac{m}{2}}$.

In the above expressions, $\overline{\gamma}_{RD}$ is the average SNR of the FSO link, $\xi_g$ represents the average power of the classical scattering component $U_S^G$, $\Omega'$ represents the average power of the coherent contribution, and $g$ is the pointing error parameter. A positive parameter in the scattering process with respect to the effective quantity of large-scale units is denoted by $\alpha$, and $\beta$ is a natural number related to the diffraction effect produced by small-scale eddies.

## 3. End-to-End SNR Statics

The end-to-end instantaneous SNR under the DF relay is expressed as:

$$\gamma_{SRD} = \frac{\gamma_{SR}\gamma_{RD}}{\gamma_{SR}+\gamma_{RD}+1} \cong \min(\gamma_{SR}, \gamma_{RD}) \tag{11}$$

The end-to-end CDF of the joint channel is

$$F_{\gamma_{SRD}}(\gamma) = P_r[\min(\gamma_{SR}, \gamma_{RD}) < \gamma] = F_{\gamma_{SR}}(\gamma) + F_{\gamma_{RD}}(\gamma) - F_{\gamma_{SR}}(\gamma)F_{\gamma_{RD}}(\gamma) \tag{12}$$

Substituting Equations (6) and (10) into Equation (12) yields:

$$
\begin{aligned}
F_{\gamma_{SRD}}(\gamma) \quad &= 1 - \exp\left(-\tfrac{m_{SR}}{\Omega_{SR}}\gamma\right) \sum_{t=0}^{m_{SR}N_S-1} \tfrac{1}{t!}\left(\tfrac{m_{SR}}{\Omega_{SR}}\gamma\right)^t \\
&+ \exp\left(-\tfrac{m_{SR}}{\Omega_{SR}}\gamma\right) \sum_{t=0}^{m_{SR}N_S-1} \tfrac{1}{t!}\left(\tfrac{m_{SR}}{\Omega_{SR}}\gamma\right)^t \tfrac{g^2 A}{2} \sum_{m=1}^{\beta} b_m G_{2,4}^{3,1}\left[\tfrac{B\gamma}{\overline{\gamma}_{RD}}\Big|\begin{matrix}1,g^2+1\\g^2,\alpha,m,0\end{matrix}\right]
\end{aligned}
\tag{13}
$$

The simple asymptotic expression of interrupt probability is derived, so as to obtain more understanding of system performance. When $\overline{\gamma}_{RD} \to \infty$, the CDF asymptotic expression of $\gamma_{RD}$ is [18]:

$$
F_{\gamma_{RD}}^{\infty}(\gamma) \approx \tfrac{g^2 A}{2} \sum_{m=1}^{\beta} b_m \sum_{k=1}^{3} \left(\tfrac{B\gamma_{RD}}{\overline{\gamma}_{RD}}\right)^{\kappa_{2,k}} \frac{\prod\limits_{l=1;l\neq k}^{3} \Gamma(\kappa_{2,l}-\kappa_{2,k})}{\kappa_{2,k}\prod\limits_{l=2}^{2}\Gamma(\kappa_{1,l}-\kappa_{2,k})}
\tag{14}
$$

Substituting Equations (6) and (14) into Equation (12), the CDF asymptotic expression can be obtained as:

$$
\begin{aligned}
F_{\gamma_{SRD}}^{\infty}(\gamma) \quad &\approx 1 - \exp\left(-\tfrac{m_{SR}}{\Omega_{SR}}\gamma\right) \sum_{t=0}^{m_{SR}N_S-1} \tfrac{1}{t!}\left(\tfrac{m_{SR}}{\Omega_{SR}}\gamma\right)^t \\
&+ \exp\left(-\tfrac{m_{SR}}{\Omega_{SR}}\gamma\right) \sum_{t=0}^{m_{SR}N_S-1} \tfrac{1}{t!}\left(\tfrac{m_{SR}}{\Omega_{SR}}\gamma\right)^t \tfrac{g^2 A}{2} \sum_{m=1}^{\beta} b_m \sum_{k=1}^{3} \left(\tfrac{B\gamma}{\overline{\gamma}_{RD}}\right)^{\kappa_{2,k}} \frac{\prod\limits_{l=1;l\neq k}^{3} \Gamma(\kappa_{2,l}-\kappa_{2,k})}{\kappa_{2,k}\prod\limits_{l=2}^{2}\Gamma(\kappa_{1,l}-\kappa_{2,k})}
\end{aligned}
\tag{15}
$$

## 4. Secrecy Outage Probability Analysis

Secrecy outage probability (SOP) is one of the secrecy benchmarks. It is defined as the occurrence of secrecy outage probability events when the instantaneous security capacity is lower than the target security rate $R_S$. Therefore, the lower bound expression of SOP for the mixed system is as follows [19]:

$$
P_{out}^{L}(R_s) = \int_0^{\infty} F_{SRD}(\theta\gamma) f_{SE}(\gamma) d\gamma
\tag{16}
$$

where $\theta = \exp(R_S)$. Substituting Equations (8) and (13) into Equation (16), the end-to-end SOP can be written as:

$$
P_{\gamma_{SRD}}(R_s) \triangleq = K_1 - K_2 + K_3
\tag{17}
$$

$$
\begin{aligned}
K_1 &= \tfrac{1}{\Gamma(m_{SE})\Gamma(m_{JE}(N_J-1))} \left[\tfrac{m_{SE}}{\overline{\gamma}_{SE}(N_J-1)}\right]^{m_E} \sum_{k=0}^{m_E}\binom{m_E}{k}\left(\tfrac{m_{JE}}{\Omega_J}\right)^{-k} \\
&\times \int_0^{\infty} \gamma^{m_{SE}-1}\exp\left[-\tfrac{m_{SE}}{\Omega_{SE}(N_J-1)}\gamma\right] G_{1,1}^{1,1}\left[\tfrac{m_{SE}\Omega_{JE}}{\Omega_{SE}m_{JE}(N_J-1)}\gamma\Big|\begin{matrix}1-k-m_{JE}(N_J-1)\\0\end{matrix}\right]d\gamma
\end{aligned}
\tag{18}
$$

$$
\begin{aligned}
K_2 &= \sum_{t=0}^{N_R m_{SR}-1} \tfrac{1}{t!}\left(\tfrac{m_{SR}\theta}{\Omega_{SR}}\right)^t \tfrac{1}{\Gamma(m_E)\Gamma(m_{JE}(N_J-1))} \left[\tfrac{m_{SE}}{\Omega_{SE}(N_J-1)}\right]^{m_{SE}} \sum_{k=0}^{m_{SE}}\binom{m_{SE}}{k}\left(\tfrac{m_{JE}}{\Omega_{JE}}\right)^k \\
&\times \int_0^{\infty} \gamma^{t+m_{SE}-1}\exp\left[-\left(\tfrac{m_{SR}\theta}{\Omega_{SR}}+\tfrac{m_{SE}}{\Omega_{SE}(N_J-1)}\right)\gamma\right] G_{1,1}^{1,1}\left[\tfrac{m_{SE}\Omega_{JE}}{\Omega_{SE}m_{JE}(N_J-1)}\gamma\Big|\begin{matrix}1-k-m_{JE}(N_J-1)\\0\end{matrix}\right]d\gamma
\end{aligned}
\tag{19}
$$

$$
\begin{aligned}
K_3 &= \sum_{t=0}^{N_R m_{SR}-1} \tfrac{1}{t!}\left(\tfrac{m_{SR}\theta}{\Omega_{SR}}\right)^t \tfrac{1}{\Gamma(m_{SE})\Gamma(m_{RE}(N_J-1))} \left[\tfrac{m_{SE}}{\Omega_{SE}(N_J-1)}\right]^{m_{SE}} \sum_{k=0}^{m_{SE}}\binom{m_{SE}}{k}\left(\tfrac{m_{JE}}{\Omega_{JE}}\right)^k \\
&\times \int_0^{\infty} \gamma^{t+m_{SE}-1}\exp\left[-\left(\tfrac{m_{SR}\theta}{\Omega_{SR}}+\tfrac{m_{SE}}{\Omega_{SE}(N_J-1)}\right)\gamma\right] G_{1,1}^{1,1}\left[\tfrac{m_{SE}\Omega_{JE}}{\Omega_{SE}m_{JE}(N_J-1)}\gamma\Big|\begin{matrix}1-k-m_{JE}(N_J-1)\\0\end{matrix}\right] \\
&\times \tfrac{g^2 A}{2} \sum_{m=1}^{\beta} b_m G_{2,4}^{3,1}\left[\tfrac{B\theta\gamma}{\overline{\gamma}_{RD}}\Big|\begin{matrix}1,1+g^2\\g^2,\alpha,m,0\end{matrix}\right]d\gamma
\end{aligned}
\tag{20}
$$

By applying the integral constancy equation provided in [20], the exponential function in above equation is converted to the Meijer-G function. Subsequently, using the integral formula of Meijer-G function given in [21], and substituting Equations (18)–(20) into Equation (17), the end-to-end SOP can be obtained after a few mathematical simplification operations as follows:

$$
\begin{aligned}
P_{\gamma_{SRD}}(R_s) = &\frac{1}{\Gamma(m_{SE})\Gamma(m_{SR}(N_J-1))} \sum_{k=0}^{m_{SE}-1} \binom{m_{SE}-1}{k}\left(\frac{\Omega_{JE}}{m_{JE}}\right)^k G_{2,1}^{1,2}\left[\frac{\Omega_{JE}}{m_{JE}}\middle|\begin{matrix}1-k-m_{JE}(N_J-1),1-m_{SE}\\0\end{matrix}\right]\\
&-\sum_{t=0}^{N_S m_{SR}-1}\frac{1}{t!}\left(\frac{m_{SR}\theta}{\Omega_{SR}}\right)^t \frac{1}{\Gamma(m_{SE})\Gamma(m_{SR}(N_J-1))}\left[\frac{m_{SE}}{\Omega_{SE}(N_J-1)}\right]^{m_{SE}} \sum_{k=0}^{m_{SE}}\binom{m_{SE}}{k}\left(\frac{\Omega_{JE}}{m_{JE}}\right)^k\\
&\times\left(\begin{matrix}\left[\frac{m_{SR}\theta(N_J-1)\Omega_{SE}+m_{SE}\Omega_{SR}}{\Omega_{SE}\Omega_{SR}(N_J-1)}\right]^{-m_{SE}-t} G_{2,1}^{1,2}\left[\frac{\Omega_{SE}\Omega_{SR}m_{SE}}{(m_{SR}\theta(N_J-1)\Omega_{SE}+m_{SE}\Omega_{SR})m_{JE}}\middle|\begin{matrix}1-k-m_{JE}(N_J-1),1-m_{SE}\\0\end{matrix}\right]\\
-\sum_{j=1}^{t} H_j\gamma_j^{t+m_{SE}-0.5}\exp\left[-\left(\frac{m_{SR}\theta}{\Omega_{SR}}+\frac{m_{SE}}{\Omega_{SE}(N_J-1)}-1\right)\gamma_j\right]\\
\times G_{1,1}^{1,1}\left[\frac{m_{SE}\Omega_{JE}}{\Omega_{SE}(N_J-1)}\gamma_j\middle|\begin{matrix}1-k-m_{JE}(N_J-1)\\0\end{matrix}\right]\frac{g^2A}{2}\sum_{m=1}^{\beta}b_m G_{2,4}^{3,1}\left[\frac{B\theta\gamma_j}{\overline{\gamma}_{RD}}\middle|\begin{matrix}1,g^2+1\\g^2,\alpha,m,0\end{matrix}\right]\end{matrix}\right)
\end{aligned}
\tag{21}
$$

where $H_j = \frac{\Gamma(n+1/2)x_j}{n!(n+1)^2[L_n^{(-1/2)}(x_j)]^2}$, and $x_j$ is the jth root of the generalized Laguerre polynomial $L_n^{(-1/2)}(x)$.

When $\overline{\gamma}_{RD}\to\infty$, the asymptotic analytical expression of the lower bound SOP can be obtained as follows by substituting Equations (8) and (15) into Equation (16):

$$
\begin{aligned}
P_{\gamma_{SRD}}^{\infty}(R_s) \approx &\frac{1}{\Gamma(m_{SE})\Gamma(m_{SR}(N_J-1))} \sum_{k=0}^{m_{SE}-1}\binom{m_{SE}-1}{k}\left(\frac{\Omega_{RE}}{m_{RE}}\right)^k G_{2,1}^{1,2}\left[\frac{\Omega_{RE}}{m_{RE}}\middle|\begin{matrix}1-k-m_{RE}(N_J-1),1-m_{SE}\\0\end{matrix}\right]\\
&-\sum_{t=0}^{N_S m_{SR}-1}\frac{1}{t!}\left(\frac{m_{SR}\theta}{\Omega_{SR}}\right)^t \frac{1}{\Gamma(m_{SE})\Gamma(m_{SR}(N_J-1))}\left[\frac{m_{SE}}{\Omega_{SE}(N_J-1)}\right]^{m_{SE}}\sum_{k=0}^{m_{SE}}\binom{m_{SE}}{k}\left(\frac{\Omega_{RE}}{m_{RE}}\right)^k\\
&\times\left(\begin{matrix}\left[\frac{m_{SR}\theta(N_J-1)\Omega_{SE}+m_{SE}\Omega_{SR}}{\Omega_{SE}\Omega_{SR}(N_J-1)}\right]^{-m_{SE}-t} G_{2,1}^{1,2}\left[\frac{\Omega_{SE}\Omega_{SR}m_{SE}}{(m_{SR}\theta(N_J-1)\Omega_{SE}+m_{SE}\Omega_{SR})m_{RE}}\middle|\begin{matrix}1-k-m_{RE}(N_J-1),1-m_{SE}\\0\end{matrix}\right]\\
-\sum_{j=1}^{t}H_j\gamma_j^{t+m_{SE}-0.5}\exp\left[-\left(\frac{m_{SR}\theta}{\Omega_{SR}}+\frac{m_{SE}}{\Omega_{SE}(N_J-1)}-1\right)\gamma_j\right]\\
\times G_{1,1}^{1,1}\left[\frac{m_{SE}\Omega_{JE}}{\Omega_{SE}(N_J-1)}\gamma_j\middle|\begin{matrix}1-k-m_{RE}(N_{RE}-1)\\0\end{matrix}\right]\frac{g^2A}{2}\sum_{m=1}^{\beta}b_m\sum_{k=1}^{3}\left(\frac{B\theta\gamma_j}{\overline{\gamma}}\right)^{\kappa_{2,k}}\frac{\prod_{l=1;l\neq k}^{3}\Gamma(\kappa_{2,l}-\kappa_{2,k})}{\kappa_{2,k}\prod_{l=2}^{2}\Gamma(\kappa_{1,l}-\kappa_{2,k})}\end{matrix}\right)
\end{aligned}
\tag{22}
$$

## 5. Average Secrecy Capacity Analysis

The average secrecy capacity (ASC) is an important indicator for assessing the security performance of active eavesdropping. It can be expressed as:

$$
\overline{C}_S = \int_0^{\infty}\frac{F_{SE}(\gamma)}{1+\gamma}(1-F_{eq}(\gamma))d\gamma
\tag{23}
$$

Substituting Equations (7) and (13) into Equation (23), a few mathematical simplifications mentioned above yields the following expression

$$
\begin{aligned}
\overline{C}_S = &\sum_{p=0}^{N_S m_{SR}-1}\frac{1}{p!}\left(\frac{m_{SR}}{\Omega_{SR}}\right)^p\left(\begin{matrix}G_{1,2}^{2,1}\left[\frac{m_{SR}}{\Omega_{SR}}\middle|\begin{matrix}-p\\0,-p\end{matrix}\right]-\sum_{j=1}^{t}H_j\frac{\gamma_j^{p+0.5}}{1+\gamma_j}\exp\left[-\left(\frac{m_{SR}}{\Omega_{SR}}-1\right)\gamma_j\right]\\\times\frac{g^2A}{2}\sum_{m=1}^{\beta}b_m G_{2,4}^{3,1}\left[\frac{B\gamma}{\overline{\gamma}_{RD}}\middle|\begin{matrix}1,g^2+1\\g^2,\alpha,m,0\end{matrix}\right]\end{matrix}\right)\\
&-\sum_{t=0}^{m_{SE}-1}\frac{1}{t!}\left[\frac{m_{SE}}{\Omega_{SE}(N_J-1)}\right]^t\frac{1}{\Gamma(m_{JE}(N_J-1))}\sum_{k=0}^{t}\binom{t}{k}\left(\frac{\Omega_{JE}}{m_{JE}}\right)^k\\
&\times\sum_{p=0}^{N_S m_{SR}-1}\frac{1}{p!}\left(\frac{m_{SR}}{\Omega_{SR}}\right)^p\sum_{j=1}^{t}H_j\frac{\gamma_j^{p+0.5}}{1+\gamma_j}\exp\left[-\left(\frac{m_{SR}}{\Omega_{SR}}+\frac{m_{SE}}{\Omega_{SE}(N_J-1)}-1\right)\gamma_j\right]\\
&\times G_{1,1}^{1,1}\left[\frac{m_{SE}\Omega_{JE}}{\Omega_{SE}(N_J-1)}\gamma_j\middle|\begin{matrix}1-k-m_{JE}(N_J-1)\\0\end{matrix}\right]\left(1-\frac{g^2A}{2}\sum_{m=1}^{\beta}b_m G_{2,4}^{3,1}\left[\frac{B\gamma_j}{\overline{\gamma}_{RD}}\middle|\begin{matrix}1,g^2+1\\g^2,\alpha,m,0\end{matrix}\right]\right)
\end{aligned}
\tag{24}
$$

## 6. Simulation Results and Analysis

In this section, the simulation results of the RF/FSO system under the influence of various parameters are provided. Furthermore, Monte Carlo simulations are used to verify the accuracy of the numerical results. The following parameters are assumed: For the RF link $d_{SE} = d_{SR} = 10$ m, $d_{RE} = 5$ m, the FSO link distance is 1 km, the wavelength is 785 nm, the optical wave number is given by $k = 2\pi/\lambda$, the refractive-index structure constant $C_n^2 = 10^{-11}$, the FSO link instantaneous SNR $\overline{\gamma}_{RD} = 20$ dB, the instantaneous SNR of the eavesdropping link $\lambda_{SE} = -10$ dB, and the target secrecy rate $R_S = 0.01$ nat/s. Other parameters are, $\xi = 0.8$, $\lambda = 0.8$, $m_{SR} = m_{SE} = 2$, $N_J = 2$, $N_R = 1$, and $\lambda_{JE} = -10$ db. In the following, the above simulation parameters values are used unless specified otherwise. When calculating the generalized Laguerre orthogonal numerical integration method, j is set to 30 to make the series converge. In order to verify the validity of the analytical expressions, Monte Carlo simulation results are provided.

Figure 3 describes the relationship between the SOP and the instantaneous SNR $\lambda_{SR}$ of the RF link in the RF/FSO system when the eavesdropper is subjected to interference corresponding to different values of SNR $\lambda_{JE}$. The simulation results show that the SOP of the system decreases gradually with the increase of $\lambda_{SR}$. When $\lambda_{SR} = 30$ dB, the system SOP values are $5.06 \times 10^{-5}$, $3.84 \times 10^{-5}$, $3.07 \times 10^{-5}$ and $2.07 \times 10^{-5}$ for $\lambda_{JE} = 2, 4, 6$ and 8, respectively. The SOP of the system is $7.38 \times 10^{-5}$ in the absence of relay interference. It can be observed that the SOP of the system decreases significantly as $\lambda_{JE}$ increases, indicating that the energy-harvesting relay can significantly improve the safety performance of the system by sending interference signals to eavesdroppers. During the communication process, increasing the SNR $\lambda_{JE}$ of interference can negatively impact the quality of the signal received by the eavesdropper; however, $\lambda_{JE}$ will be limited by the transmitted power. Therefore, introducing the energy-harvesting relay interference and selecting a reasonable transmission power can effectively improve the security performance of the system.

Figure 4 shows the relationship between SOP of the mixed RF/FSO system and the instantaneous SNR $\lambda_{SR}$ of the RF link under different number of jamming antennas $N_J$ at the energy-harvesting relay end. It can be observed from the simulation results that, when $\lambda_{SR} = 30$ dB, the SOP values of the system are $5.06 \times 10^{-5}$, $1.44 \times 10^{-5}$, $4.02 \times 10^{-6}$ and $1.36 \times 10^{-6}$ for the number of interferer antennas $N_J = 2, 4, 6$ and 8, respectively. This shows that the security performance of the system can be improved by increasing the number of jamming transmitting antennas. Limited by cost and physical space, the number of interferer antennas cannot be increased indefinitely. This behavior is caused due to the increasing of spatial freedom of the relay node with the increase of the number of antennas, which can reduce the communication quality of the eavesdropping channel according to its status information. The transmission power of the transmitter is lower under the scheme with a higher number of jamming antennas for the same value of SOP. This provides a good solution for designing the jammer equipment and reducing energy consumption at the transmitter side.

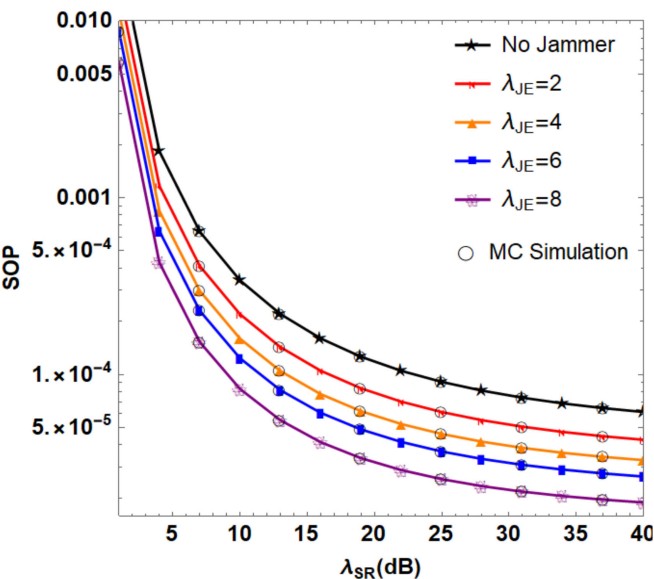

**Figure 3.** Simulation diagram of SOP under different SNR $\lambda_{JE}$ of interference in RF/FSO system.

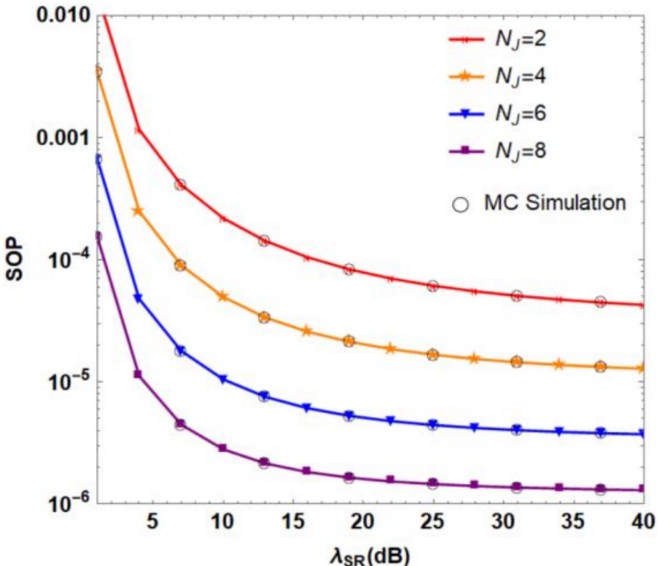

**Figure 4.** Simulation diagram of SOP under different number of antennas $N_J$ in RF/FSO system.

Figure 5 describes the relationship between the SOP of the RF/FSO system and the instantaneous SNR $\lambda_{SR}$ of the RF link under different values of energy conversion efficiency $\zeta$ of the energy-harvesting relay. It can be noted from the figure that the SOP of the system decreases with the increase of $\lambda_{SR}$. When $\lambda_{SR} = 30$ dB, the SOP values of the system are $5.06 \times 10^{-5}$, $4.18 \times 10^{-5}$, $3.55 \times 10^{-5}$ and $3.07 \times 10^{-5}$ for $\zeta = 0.3, 0.5, 0.7, 0.9$, respectively. It is known that the energy-harvesting relay converts the signals sent by the received source into a DC signal with a higher efficiency, which increases the energy collected by the relay. This subsequently improves the interference effect of the energy-harvesting relay on eavesdroppers, consequently reducing the communication quality of eavesdroppers, and improving the security of the system. The improved energy conversion efficiency scheme is adopted to achieve a more stable energy collection and provide the system with better confidentiality.

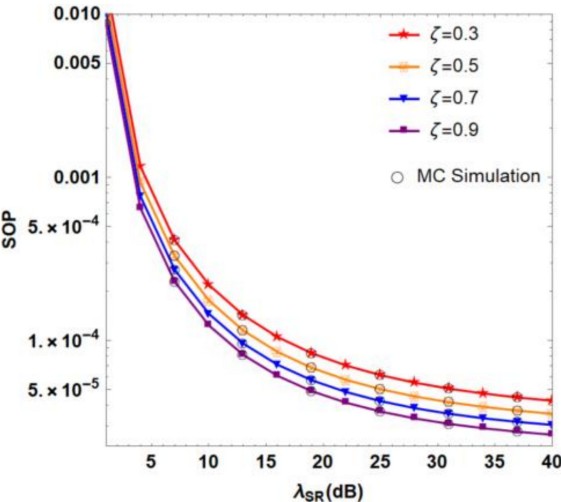

**Figure 5.** Simulation diagram of SOP under different energy conversion efficiency $\zeta$ in RF/FSO system.

Figure 6 describes the relationship between the SOP of the mixed RF/FSO system and the instantaneous SNR $\overline{\gamma}_{RD}$ of the FSO link under the influence of strong, medium and weak turbulence. As the figure shows, the curve as a whole decreases monotonically as the instantaneous SNR of the FSO link increases, and the error plane appears in the high SNR region, which is consistent with the results of asymptotic analysis. When $\overline{\gamma}_{RD} = 30$ dB, the SOP values of the system under weak, medium and strong turbulence are $1.9 \times 10^{-3}$, $1.3 \times 10^{-3}$ and $1.2 \times 10^{-3}$, respectively. Comparing the simulation results under moderate and weak turbulence conditions, the SOP of the system under severe weather conditions is clearly higher than those under medium and weak turbulence conditions. Thus, the safety performance of the system under weak turbulence conditions is clearly better than that under strong and medium turbulence conditions. Therefore, the safety performance of the RF/FSO system is considerably affected under severe weather conditions.

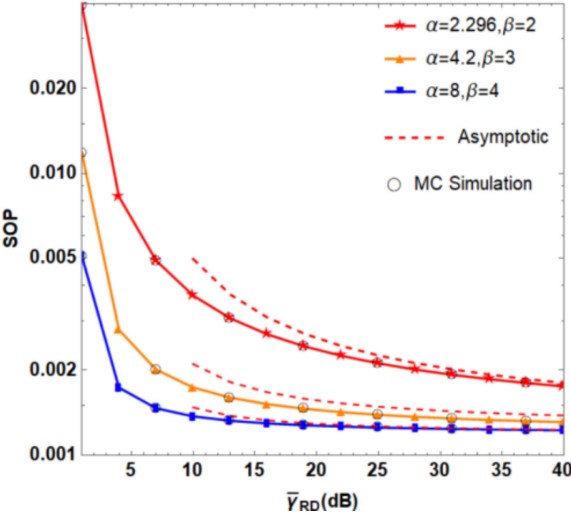

**Figure 6.** Simulation diagram of SOP with strong, medium and weak turbulence in RF/FSO system.

Figure 7 shows the relationship between the ASC of the RF/FSO system and the instantaneous SNR $\lambda_{SR}$ of the RF link under different values of power distribution factor $\lambda$ for the energy-harvesting relay. When $\lambda_{SR} = 30$ dB, the ASC values are 1.82, 1.91, 2.03 and 2.11 for $\lambda = 0.1, 0.3, 0.6$ and 0.9, respectively. It can be noted that the ASC of the system increases with the increase of $\lambda$. The power distribution factor $\lambda$ is positively correlated with the effective energy obtained by the energy-harvesting relay, which means

that the larger the energy, the higher the amount of stored energy. Thus, a higher amount of energy is used by the energy-harvesting relay to transmit interference signals, consequently improving the interference effect of the energy-harvesting relay on eavesdroppers and making the communication of the system more secure.

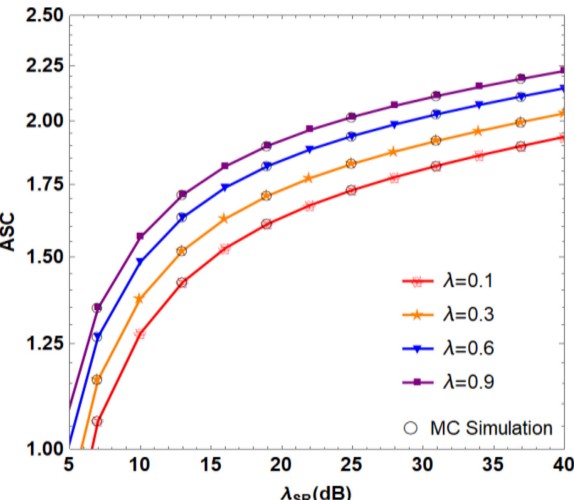

**Figure 7.** Simulation diagram of ASC under different power distribution factor $\lambda$ in RF/FSO system.

Figure 8 describes the relationship between the ASC and the instantaneous SNR $\lambda_{SR}$ of RF link in the RF/FSO system under different SNR $\lambda_{JE}$ values of the interference signal. When $\lambda_{SR} = 30$ dB, the ASC values are 2.21, 2.32, 2.37 and 2.41 for $\lambda_{JE} = 2, 4, 6$ and 9, respectively. In the absence of relay interference, the corresponding ASC is equal to 1.96. It can be observed that the ASC of the system increases to some extent with the increase of $\lambda_{JE}$. It is also evident that the energy-harvesting relay can improve the security of the system by sending interference signals to eavesdroppers and increasing the transmission power can improve the PLS of the system.

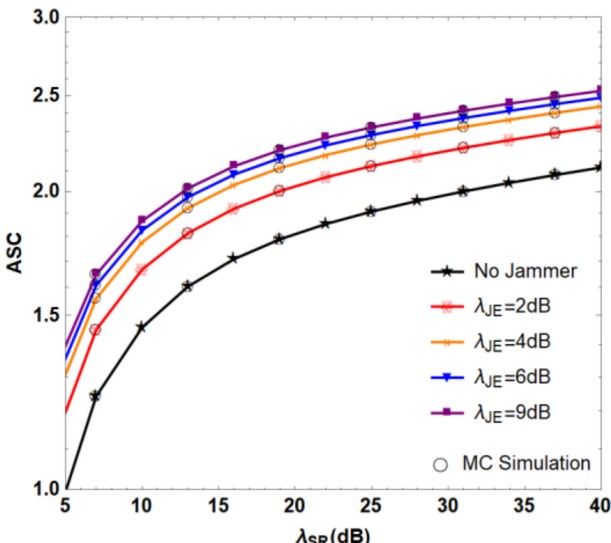

**Figure 8.** Simulation diagram of ASC under different f SNR $\lambda_{JE}$ of interference in RF/FSO system.

## 7. Conclusions

In this paper, the safety performance of SWIPT energy-harvesting relay jamming based mixed RF/FSO systems was studied. The safety outage probability (SOP) and average security capacity (ASC) of the system were analyzed by theoretical derivation

and simulation, and the validity of the expression was verified using the Monte Carlo methods. The effects of the signal-interference noise ratio of the energy-harvesting relay, number of relay interferer antennas, power distribution factor, energy conversion efficiency and atmospheric turbulence on system safety performance were studied. The simulation results showed that the SOP of the system decreased and the ASC increased as the instantaneous SNR of the RF link increased. When the signal-interference noise ratio of the energy-harvesting relay was increased, the SOP and ASC of the system were clearly decreased. This behavior indicated that the interference effect on eavesdroppers could be improved by increasing the signal-interference noise ratio, which enhanced the security performance of the system. When the number of interferer antennas increased, the SOP of the system decreased significantly. For the same value of SOP, the transmitter power could be reduced by increasing the number of interferer antennas, which provided a good scheme for the jammer equipment to reduce the energy consumption of the transmitter. The simultaneous wireless information and power transfer (SWIPT) technology was used to adjust the power distribution factor to increase the power received by the energy collector, which strengthened the interference signal transmitted by the energy-harvesting relay, and improved the security of the system. On the other hand, the increase of energy conversion efficiency enabled the energy collector to store more energy, enhance the interference signal during signal transmission, render the collection process controllable, and improve the security capability of the system. In addition, under different turbulence conditions, the SOP of the system decreased as the instantaneous SNR of the FSO link increased. When the atmospheric turbulence became weak, the SOP of the system decreased, indicating that the improvement of weather conditions could improve the physical layer security of the system. In conclusion, the safety performance of the RF/FSO system improved significantly under the action of the energy-harvesting relay jamming based on SWIPT, which could provide a good theoretical basis for engineering implementation.

**Author Contributions:** Methodology, Z.Z. and Z.S.; software, Z.Z.; validation, Z.Z. and Z.S.; formal analysis, Y.W.; writing—original draft preparation, Z.Z.; writing—review and editing, Z.Z. and Y.W.; supervision, Y.W. All authors have read and agreed to the published version of the manuscript.

**Funding:** This research was funded by the Research Fund of State Key Laboratory of Coal Resources and safe Mining, CUMT (SKLCRSM21KF010); the National Natural Science Foundation of China (No. 51704267).

**Data Availability Statement:** Not applicable.

**Conflicts of Interest:** The authors declare no conflict of interest.

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
