# Peer review of "On Secrecy Performance of SWIPT Energy-Harvesting Relay Jamming Based Mixed RF-FSO Systems"

_photonics, doi:10.3390/photonics9060374_

Round 1
Reviewer 1 Report
The manuscript reports a simultaneous wireless information and power transfer (SWIPT) energy-harvesting relay jamming based mixed RF/FSO system. The authors analyzed the safety outage probability (SOP) and average security capacity (ASC) of the system . The manuscript should be published in Photonics after authors amend some minor items listed below.
1. The label text in Figure 2 should be changed from Chinese to English. Figure 2 is too simple. Authors should optimize this figure to show the SWIPT power switch receiver structure more clearly.
2. Limited by cost and physical space, the number of interferer antennas cannot be increased indefinitely. The author needs to point out this limitation in the analysis of the simulation results in Figure 4.
3. The change of SOP under different energy conversion efficiency ζ is not obivous in Figure 5. The scale of SOP in Figure 5 should be adjusted to make the change more obvious.
Reviewer 2 Report
This paper investigates the security performance of SWIPT energy-harvesting relay jamming based mixed RF-FSO systems. The closed expressions for the secrecy outage probability (SOP) and the average secrecy capacity (ASC) of the mixed RF-FSO system are derived. The simulation results presented in the paper may raise interest to the research community in the field. In my opinion, this manuscript can be accepted after minor revisions. My comments/questions are listed as follows.
1) So far, the RF-FSO systems have been studied considering the impact of SIMO, MIMO, TAS respectively. What are the differences between these schemes and SWIPT energy-harvesting relay jamming and what are the benefits?
2)In SWIPT structure,Why do the paper choose power switching (PS) structure?
3) Please explain why the jamming signal sent by the energy-harvesting relay only has an effect on the eavesdropper, but not on the receiver?
4) As can be seen in FIG.3, increasing the signal-interference noise ratio improves the safety performance more obviously, so why not directly improve the performance from this point?
Reviewer 3 Report
This paper proposed the studies of security performance optimization for SWIFT energy-harvesting relay jamming based mixed RF/FSO systems. This work is interesting since the numerical results were proposed and then the Monte-Carlo simulation was employed to validate the proposed results. The study investigated several parameters that affect the safety performance of the system. With the understanding of each parameter, the system can be evaluated and the performance can be improved accordingly. This paper is in a good term for publication with a small suggestion as:
- In Fig.2, English translation should be used.
- In Fig. 3 caption, there’s extra ‘f’ in front of SNR.
Reviewer 4 Report
The authors proposed a simultaneous wireless information and power transfer energy-harvesting relay jamming based mixed RF/FSO system, and studied its security performance optimization in the presence of an eavesdropper. The results are interesting, I feel that the manuscript may be accepted for this Journal after the minor revisions.
1. The authors can provide the expressions of the Meijer-G function in the manuscript.
2. What is the meanings of the parameters of hsr and T in the Eq.2, hRE in the Eq.3?
